# Microstructure Evolution during High-Pressure Torsion in a 7xxx AlZnMgZr Alloy

**DOI:** 10.3390/ma17030585

**Published:** 2024-01-25

**Authors:** Anwar Qasim Ahmed, Dániel Olasz, Elena V. Bobruk, Ruslan Z. Valiev, Nguyen Q. Chinh

**Affiliations:** 1Department of Materials Physics, ELTE Eötvös Loránd University, Pázmány Péter Sétány 1/A, 1117 Budapest, Hungary; anwar.ahmed.qasim@gmail.com (A.Q.A.);; 2College of Science, University of Kufa, Najaf 54001, Iraq; 3Institute for Technical Physics and Materials Science, Centre for Energy Research, Budapest Konkoly-Thege út 29-33, 1121 Budapest, Hungary; 4Institute of Physics of Advanced Materials, Ufa University of Science and Technology, 32 Zaki Validi Street., Ufa 450076, Russia; e-bobruk@yandex.ru (E.V.B.); ruslan.valiev@ugatu.su (R.Z.V.); 5Laboratory for Dynamics and Extreme Performance of Advanced Nanostructured Materials, Saint Petersburg State University, 28 Universitetsky Prospekt, Peterhof, St. Petersburg 198504, Russia

**Keywords:** AlZnMg alloy, HPT, UFG, TEM, DSC, hardness, tensile test, decomposition, GB segregation, superplasticity

## Abstract

A homogenized, supersaturated AlZnMgZr alloy was processed via severe plastic deformation (SPD) using a high-pressure torsion (HPT) technique for different revolutions at room temperature to obtain an ultrafine-grained (UFG) microstructure. The microstructure and mechanical properties of the UFG samples were then studied using transmission electron microscopy (TEM), differential scanning calorimetry (DSC), and tensile and hardness measurements. The main purpose was to study the effect of shear strain on the evolution of the microstructure of the investigated alloy. We found a very interesting evolution of the decomposed microstructure in a wide range of shear strains imposed by HPT. While the global properties, such as the average grain size (~200 nm) and hardness (~2200 MPa) appeared unchanged, the local microstructure was continuously transformed. After 1 turn of HPT, the decomposed UFG structure contained relatively large precipitates inside grains. In the sample processed by five turns in HPT, the segregation of Zn atoms into grain boundaries (GBs) was also observed. After 10 turns, more Zn atoms were segregated into GBs and only smaller-sized precipitates were observed inside grains. The intensive solute segregations into GBs may significantly affect the ductility of the material, leading to its ultralow-temperature superplasticity. Our findings pave the way for achieving advanced microstructural and mechanical properties in nanostructured metals and alloys by engineering their precipitation and segregation by means of applying different HPT regimes.

## 1. Introduction

In recent decades, exploring materials with high corrosion resistance and excellent strength-to-weight ratios has become a priority in the automotive and aircraft industries to improve load capacity and fuel efficiency. Accordingly, shipbuilders and carmakers have turned toward aluminum alloys, which are considered a viable alternative to the heavier steel currently adopted. The properties of Al alloys are improved to meet the requirements of these industries by subjecting Al to severe plastic deformation (SPD) [1,2,3,4]. Hence, many researchers in the material physics field have devoted their efforts to study the effect of SPD on the mechanical and microstructure properties of the Al alloys. Among the numerous SPD processes, high-pressure torsion (HPT) [5,6] is a considerable and widely applied method for investigating the significance of SPD on metallic alloys. Enormous shear strain of this technique produces an ultrafine-grained (UFG) structure and may result in the formation of nanoprecipitates in the processed materials [7,8,9,10,11]. In this regard, the mechanical and microstructural characteristics of UFG materials processed via HPT are also affected by the dissolving and precipitating of nanosized-dispersoids, in addition to the effect of the density of defects [12,13,14]. It is also well known that a significant amount of shear strain in HPT may cause the dynamic recovery and recrystallization of precipitates [15,16,17,18,19]. The effect of this phenomenon has previously been studied in many works [20,21,22,23]. Previous studies for the dynamic precipitation behavior in the HPT-processed alloy demonstrate that a decomposed solid solution at RT creates nanoscale solute structures in the matrix associated with the segregation of solute atoms at the GBs. These structures work in conjunction with the grain refinement effect and high dislocation density to improve the strength of HPT-processed alloy by suppressing grain growth during dynamic recrystallization [24]. Moreover, HPT has significant potential to produce non-equilibrium GB microstructures that have a relatively high grain boundary energy, thereby improving the segregation at GBs during the SPD process [10,11,19]. These findings demonstrate that the SPD technique is a magnificent method for designing a UFG microstructure of an Al alloy by driving grain refinement, dynamic reversion, and solute heterogeneity precipitations. 

It is well known that the AlZnMg chemical composition (7xxx) alloys have been considered as the most significant basic materials in the aluminum industries. These materials are widely investigated because of their excellent mechanical properties [11,12,13,14,15,16,17,18,19,20,21,22,23,24,25], as the initially supersaturated microstructure can be varied conveniently by applying different heat treatments and plastic deformation techniques. 

Despite extensive investigations [26,27,28,29,30], the effect of the shear strain in a wide range of deformation imposed in HPT processes on the microstructure and on the subsequent mechanical properties of several AlZnMg alloys has not been clarified in detail.

The aim of this work is to investigate the microstructure evolution of an Al-4.8%Zn-1.2%Mg-0.14%Zr (wt.%) alloy processed via HPT and its effect on mechanical properties by using transmission electron microscopy (TEM), depth-sensing indentation (DSI) and differential scanning calorimetry (DSC), as well as microhardness and tensile measurements. 

## 2. Materials and Methods

The alloy with a weight percent composition of Al-4.8Zn-1.2Mg-0.14Zr was first cast, followed by homogenization at 470 °C for 8 h and hot extrusion at 380 °C. In the next step, disks 1.4 mm thick and 20 mm in diameter were cut from the extruded rods and processed via high-pressure torsion. Prior to HPT processing, the disk samples were again homogenized at 470 °C for 1 h and quenched into water at room temperature (RT). The disks were processed to N = 1, 2, 5 or 10 revolutions (turns) of HPT under a pressure of 6 GPa at room temperature. Samples taken at the half-radius of the HPT disks were used for further investigations, where the samples will be identified as HPT-1, HPT-2, HPT-5, and HPT-10, according to the number of revolutions in the HPT process. Details of the HPT process can be found elsewhere [5,6].

The microstructure of the samples was investigated a using aTitan Themis G2 200 scanning transmission electron microscope (STEM) (Thermo Fisher Scientific, Waltham, MA, USA) operating at 200 kV was used for TEM and energy-disperse X-ray spectroscopy (EDS) investigations. Thin foils were prepared via mechanical polishing and thinned until perforation at −20 °C using usual twin-jet electropolishing, operating with a chemical solution of 33% HNO_3_ and 67% CH_3_OH. The STEM images were taken using a high-angle annular dark-field (HAADF) detector. Some experimental results on the microstructure of the HPT-10 sample have already been published [31]. In order to identify the thermal events, a differential scanning calorimeter of PerkinElmer-type equipment was used at heating rates of 10, 20, 30, and 40 K/min.

Mechanical properties of the samples were assessed by performing tensile and microhardness tests. The tensile measurements were conducted at a testing temperature of 170 °C and strain rate, $\dot{\varepsilon }$ of 5 × 10^−4^ s^−1^, by using an Instron 5982 machine. Microhardness tests were carried out through depth-sensing indentation using a Vickers indenter for maximum loads of 50 mN and applying the well-known Oliver–Pharr method [32,33].

## 3. Results

### 3.1. Effect of HPT on Mechanical Properties of AlZnMgZr Samples

Figure 1 shows the Vickers microhardness values, HV, of the investigated AlZnMgZr samples after different number of turns (N) of HPT processing. For the purpose of comparison, the hardness of the initial (Q) sample and the hardness of the undeformed sample stored at RT (naturally aged) for a long time, can also be seen. In this figure, the amount of shear strain at a certain number of HPT revolutions is also plotted.

The experimental results indicate well the strengthening effect of HPT processing. Already after the first turn, the hardness of the sample reaches 2200 MPa, which is two and a half times higher than that of the freshly annealed sample (~830 MPa), and approximately 30% higher than the hardness (~1700 MPa) of a sample naturally aged for a long time. Furthermore, it can be seen that the hardness of the HPT-processed samples is practically the same, regardless of the number of revolutions.

Figure 2 presents the creep behaviors of the different HPT-processed samples which were deformed by means of tensile tests performed at a strain rate of $5\times 10^{-4}s^{-1}$ and a testing temperature of 170 °C. Even though their hardness at room temperature is almost the same, their creep properties are significantly different and depend on the number of revolutions. It can be observed that while the maximum flow stress of the samples decreases (from 180 to 58 MPa), their maximum elongation increases (from 40 to 500%) with the increasing number of the revolutions in HPT, showing the superplasticity of the sample processed via HPT in 10 turns. It should be noted that the testing temperature of 170 °C is only 0.47 *T_m_*, lower than half of the absolute melting point (*T_m_*) of aluminum. The low-temperature superplasticity of the ultrafine-grained HPT-10 sample with a record elongation of 500% has been previously investigated in more detail [11].

### 3.2. Effect of HPT on Microstructure Observed Using TEM

The HPT processing significantly modified the microstructure of the initially coarse-grained, supersaturated sample, where only Al_3_Zr particles and Guinier–Preston (GP) zones can be found [10]. It is a well-known fact that the supersaturation of AlZnMg alloys usually leads to the formation of metastable GP zones and/or of stable η-phase (MgZn_2_ composition) precipitates [31,34,35]. While in conventional (only annealed) samples, the η-phase particles can only be observed above room temperature (usually above 80 °C); in the SPD-processed samples, these precipitates can form even at room temperature [31]. Figure 3 shows the TEM images in HAADF mode on the HPT-1 (Figure 3a) and HPT-5 (Figure 3b) samples. Figure 4 shows the microstructure of the HPT-10 sample with more details. 

Experimental results show very clearly the grain-refining effect of the HPT process, which resulted in ultrafine-grained (UFG) structures having an average grain size of about 200 nm in all investigated HPT-processed samples, leading to an increase in the hardness of these HPT-processed samples, as shown in Figure 1. Further careful investigations have shown that η-phase MgZn_2_ precipitates can also be observed in the HPT-processed samples. These precipitates appear as bright areas on the HAADF-STEM images since the atomic number of Zn is much higher than that of Al. It can also be observed that the formation of the η-phase MgZn2 particles is significantly affected by the number of revolutions (the amount of shear strain) applied during the HPT process. After one revolution (N = 1), η-phase particles between 5 and 30 nm (see Figure 3a) in size are formed. After five revolutions, more and somewhat larger particles can be observed in the size range of 20–40 nm (Figure 3b). It is interesting that after more severe shear strain in 10 revolutions in HPT, particles larger than 10–15 nm are rarely visible (see Figure 4a); rather, only much finer 2–5 nm particles can be observed (see the magnified TEM image of Figure 4b).

Given that the η-phase MgZn2 precipitates were mainly formed along or near the grain boundaries, this result suggests that during further deformation, the particles are fragmented and/or partially dissolved into the matrix, changing the structure of the grain boundaries.

Figure 5 shows the HAADF image of a typical grain boundary (Figure 5a), together with corresponding EDS element maps (Figure 5b–d) and EDS line profile analysis (Figure 5e,f) observed in the HPT-10 sample. Experimental results indicate both the relative decrease in Al (Figure 5b) and the relative increase in Mg and Zn solutes (Figure 5c–f) in the grain boundaries. The results of EDS measurements have shown that most of the grain boundaries in the HPT-10 sample can be regarded as Zn/Mg-rich boundaries. In the case of the HPT-5 sample, only a small fraction of the grain boundaries is segregated by Zn and Mg. The HPT-1 sample is not at all characterized by the presence of grain boundaries segregated by Zn and Mg solute atoms.

### 3.3. Characterization of Different HPT-Processed Microstructures Using DSC Measurements

Figure 6 shows the DSC thermograms (heat flow–temperature curves) obtained at different heating rates (V) on the different (HPT-1, HPT-2, HPT-5, and HPT-10) samples in the testing temperature range from 300 to 760 K. In all cases, the thermograms can be characterized by the presence of three peaks. First, an endothermic peak is observed during the heating process, indicating the dissolution of existing phases in the UFG microstructure. Then, an exothermic peak is followed, which is connected to the precipitation and coarsening of precipitates. The third peak is another endothermic one observed at a higher temperature range, indicating the final dissolution of precipitates formed through the whole heat treatment [36,37,38]. In the present work, where the focus is on the evolution of the microstructure during HPT processing, the characteristics of the first two (endo- and exothermic) are primarily investigated and analyzed.

Using the experimentally measured DSC thermograms, the specific heats (enthalpies) of the dissolution (for endothermic reaction), ∆*H_d_* and that of precipitation (for exothermic reaction), ∆*H_p_* can be determined. Experimental results have shown that for a given sample, these quantities obtained at different heating rates are almost the same. The quantities obtained for different samples indicate how the fraction of dissolved or formed phases change in these samples [10,39,40]. In the present work, the measured specific heats are listed in Table 1, showing well the tendency of the microstructural evolution during the HPT process.

#### 3.3.1. The Activation Energy of Transformation Processes

It is well known that the transformation dynamic requires activation energy to diffuse their constituents of solutes in the microstructure through the reversion and precipitation processes. This energy value is changed greatly by modifying the microstructure via the SPD process. Because of thermal activation, the peak temperature, *T_p_*, of both reactions (endothermic and exothermic) shifts toward higher temperatures when increasing the heating rate during DSC measurements. The activation energies for these reactions can be determined by using the well-known Kissinger equation [41,42]: (1)$$\ln\left(\frac{V}{T_{p}^{2}}\right)=C+\frac{Q}{RT_{p}},$$
where *V* is the heating rate, *T_p_* represents the peak temperature of a given reaction, *C* is a material constant, and *R* is the universal gas constant. In the following, the activation energies are denoted as *Q_d_* and *Q_p_* for the dissolution and precipitation reactions, respectively.

Figure 7 shows the Kissinger plots obtained for the first dissolution (Figure 7a) and precipitation (Figure 7b) reactions in the investigated HPT-processed samples. In all cases, the data points can be fitted well with a linear line. According to Equation (1), the activation energies for different processes can be determined from the slopes of the fitted lines, within 10% relative errors. During the exothermic reaction of the HPT-10 sample, the data were more scattered, and the relative error was nearly 20%. The obtained values can also be seen in the figures.

Experimental results show that both activation energies are affected significantly by the number of HPT revolutions. By increasing the number of revolutions from 1 to 10, the activation energy characterizing the dissolution reaction (see Figure 7a) monotonously decreases from 108 to only 50 kJ/mol. At the same time, the activation energy value of the precipitation process (Figure 7b) slightly decreases from 49 to 26 kJ/mol. 

#### 3.3.2. Kinetic Parameters for Different Reactions during DSC Measurements

It is well established that the kinetics of the dissolution and precipitation reactions that take place in the materials when imposed to DSC can be characterized by the kinetics parameters developed using the Avrami–Johnson–Mehl theory [42,43]. These quantities are represented by the transformed volume fraction, *Y,* given by:(2)$$Y\left(T\right)=\frac{A\left(T\right)}{A(T_{f})}$$
and the transformation rate, *dY/dt*, which can be given as:(3)$$\frac{dY}{dt}=\frac{dY}{dT}\cdot \frac{dT}{dt}=\frac{dY}{dT}\cdot V$$
at the heating rate *V*. The quantity A(T) is the area under the given endo- or exothermic peak in the temperature range from the initial temperature, $T_{i}$, to actual temperature, T. *T_f_* is the final temperature of this peak.

Figure 8 shows the Y–T connections characterizing the dissolution (Figure 8a–c) and precipitation (Figure 8d–f) in HPT-1, HPT-5, and HPT-10 samples measured at different heating rates. These sigmodal-shaped functions shift to higher temperatures as the shear strain increases by increasing the HPT revolution number and at increasing heating rates. The temperature range covering the endothermic reaction for the HPT-1 and HPT-5 samples appears slightly different. It is extended from around 390 to 457 K, i.e., extending over 67 K. Further, for the HPT-10 sample, this interval begins earlier, at 380 K, and finishes later, at 471 K, extending over 91 K, which is one and a half times larger than that for the HPT-1 and HPT-5 samples. This behavior indicates that the dissolved precipitates in the HPT-10 samples are less stable, so they dissolved earlier into the matrix. Furthermore, the beginning of the exothermic reaction in the HPT-10 sample also shifted slightly toward a lower temperature. It can be observed that the precipitation reaction started at 434 K and 420 K for the HPT-1 and HPT-10 samples, respectively, indicating the easier precipitation in the HPT-10 sample.

Figure 9 shows the change of the transformation rate, *dY/dt*, in the function of temperature for the HPT-1, HPT-5, and HPT-10 samples. It can be seen that the transformation rate is low at the beginning and at the end of the given process, and there is a rapid change in between, showing a maximum located in the middle region. In the case of endothermic reactions (Figure 9a–c), the highest transformation rate for the HPT-1 sample is located at 423 K, whereas it is located at 416 K for the HPT-10 sample, i.e., there is a shifting toward lower temperatures. In addition, the kinetics reversion in the microstructure of the HPT-1 sample occurred at a higher rate compared with the HPT-10 one, as the maximum transformation rates of these two samples are 16 × 10^−3^ and 12 × 10^−3^ s^−1^, respectively. This behavior is ascribed to the dissolved precipitations of smaller size, which caused them to dissolve more quickly and easily in the case of the HPT-10 sample. Additionally, for the exothermic reaction, the transformation process in the HPT-1 sample occurs at a lower rate (12 × 10^−3^ s^−1^) compared with that of the HPT-5 (14 × 10^−3^ s^−1^) and HPT-10 (18 × 10^−3^ s^−1^) samples, as shown in Figure 9d–f.

## 4. Discussion

The results of the TEM investigations show that the grain-refining effect of the HPT process resulted in stable ultrafine-grained structures with an average grain size of about 200 nm in all HPT-processed samples. The strength of HPT-processed samples is primarily determined by the UFG structure via the Hall–Petch effect [44,45], leading to almost the same hardness observed in these HPT-processed samples, as shown in Figure 1. Besides the grain-refining effect, the HPT processing also resulted in the decomposition of the initially supersaturated microstructure. When increasing the number of HPT revolutions from 1 to 10, the decomposed microstructure significantly changed from a structure containing only relatively large MgZn_2_ η-phase particles to one containing both smaller η-phase particles and a high fraction of grain boundaries (GBs) segregated by solute atoms, mainly Zn. Due to the effect of solute-segregated GBs, the creep behavior of the investigated HPT-processed samples deformed through tensile tests at 170 °C strongly depends on the number of HPT revolutions. While the HPT-1 sample shows a very poor elongation lower than 50%, the HPT-10 one can be deformed in terms of superplasticity, with an elongation higher than 500% under the same conditions.

Results of DSC measurements confirmed the significant difference between the precipitate structures formed in the HPT-10 and those formed in the other samples processed with fewer revolutions. The relatively high activation energy (*Q_d_*) values obtained for the dissolution (endothermic) reactions in HPT-1 and HPT-2 (see Figure 7a) after few revolutions in HPT processing are related to the fragmentation of undissolved coarse η phase due to the abundance of this phase because these samples were applied to low-strain energy. On the contrary, the relatively high 10 revolutions led to the dissolution of this fragmented intermetallic compounds of the η phase into the Al matrix in the HPT-10 sample due to the high deformation energy. The low activation energy (*Q_d_*) value characterizing the dissolution (endothermic) reactions in the HPT-10 sample is caused mainly by the dissolution of the low fraction of η phase, which remained at grain boundaries because the dissolution of precipitates inside the grains is less pronounced [28]. Following the dissolution reaction at the first endothermic process, the formation/re-formation of precipitates occurred as presented by the exothermic reaction peak in the DSC line profile. This process can also be characterized by the decreasing activation energy (*Q_p_*) at increasing numbers of HPT revolutions, as shown in Figure 7b. This is the consequence of the increasing volume fraction of GBs that facilitates the nucleation of precipitations by presenting surfaces of lower inter-phase boundary energy and of increasing dislocation density, which promote the diffusion process [46,47].

Considering the experimental results obtained through mechanical, TEM, and DSC measurements, a schematic representation of the microstructural development of the 7xxx series AlZnMg alloys during the HPT process can be suggested through the main steps shown in Figure 10. It is well established that in the early stage of the HPT processing, the dislocation density increases due to the intensive plastic deformation and the pinning effect of the solute atoms in the initially supersaturated microstructure. The strong increase in dislocation density leads to the formation of a UFG structure, and to the decomposition of the supersaturated structure via the formation of relatively large η-phase MgZn2 particles near grain boundaries. While the UFG structure remains stable in a wide range of shear strain in HPT from one to ten revolutions, the decomposed structure is continuously changing. Due to the effect of further shear strain, the formed η-phase particles are fragmented and/or partially dissolved back into the matrix or segregated into the grain boundaries, significantly changing the creep behavior of the UFG alloy.

## 5. Conclusions

The microstructural and mechanical properties of an HPT-processed AlZnMgZr alloy were studied via transmission electron microscopy, differential scanning calorimetry, and tensile and hardness measurements. Emphasis was placed on the effect of shear strain on the evolution of the microstructure of the investigated alloy. The main results can be summarized as follows:(1)Severe plastic deformation exerted via HPT resulted in a stable ultrafine-grained structure with a grain size of about 200 nm, significantly increasing the strength of the material at room temperature.(2)Together with the formation of the UFG structure, the HPT process also resulted in the decomposition of the initially supersaturated structure through the formation of η-phase MgZn2 particles near grain boundaries in the investigated AlZnMgZr alloy.(3)While the UFG structure remains stable during the HPT process, the decomposed structure is continuously changing. Due to the effect of further shear strain, the formed η-phase particles are fragmented and/or partially dissolved back into the matrix or segregated into the grain boundaries, significantly changing the creep behavior of the UFG alloy.(4)The intensive segregation of solute atoms into GBs may have an important role in the creep behavior of the material, leading to its ultralow-temperature superplasticity. The obtained results pave the way for achieving advanced microstructural and mechanical properties in nanostructured metals and alloys by engineering their precipitation and segregation by means of applying different number of HPT revolutions.

## Figures and Tables

**Figure 1 materials-17-00585-f001:**
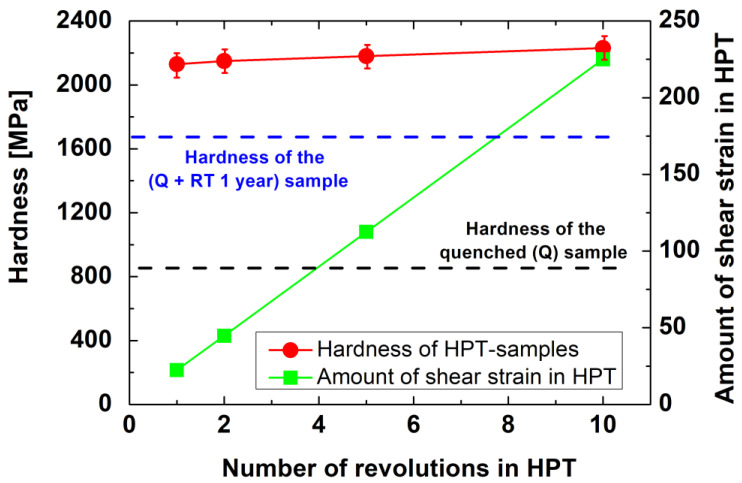
Microhardness of the investigated HPT-processed AlZnMgZr samples.

**Figure 2 materials-17-00585-f002:**
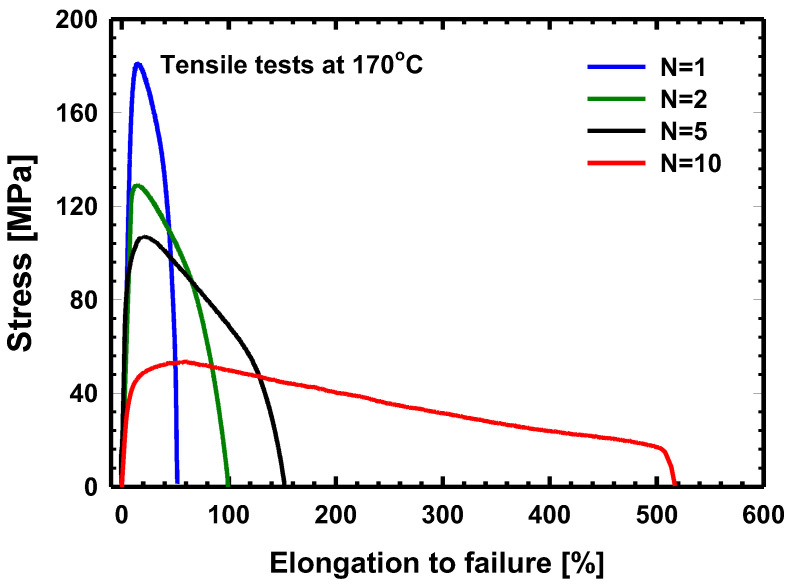
Stress–elongation curves of the samples processed via HPT in different (N) revolutions and deformed via tensile test.

**Figure 3 materials-17-00585-f003:**
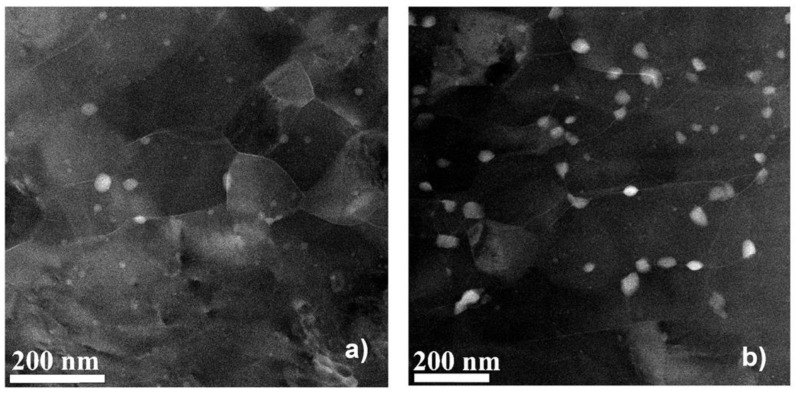
STEM-HAADF images of the microstructures of (**a**) HPT-1 and (**b**) HPT-5 samples.

**Figure 4 materials-17-00585-f004:**
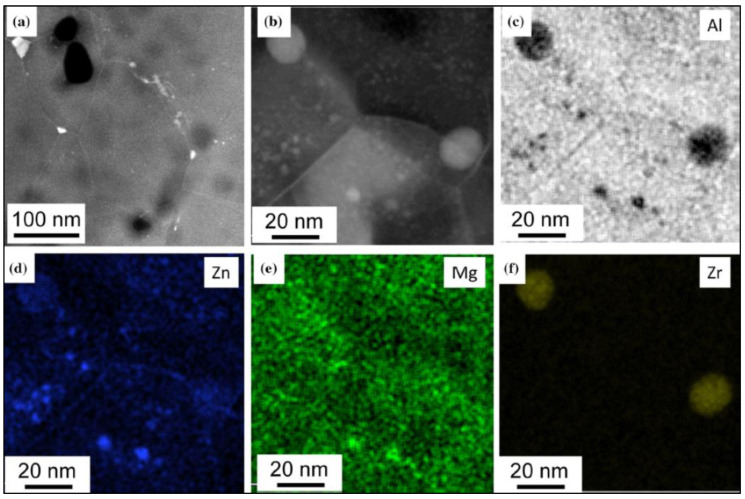
UFG microstructure of the investigated HPT-10 sample taken as (**a**,**b**) HAADF STEM images in low and higher magnifications, respectively, and (**c**–**f**) corresponding EDS elemental maps obtained on the area shown in (**b**), indicating the presence of small Mg/Zn precipitates together with two Al_3_Zr particles. Reproduced from Ref. [31]. Copyright 2020, Springer.

**Figure 5 materials-17-00585-f005:**
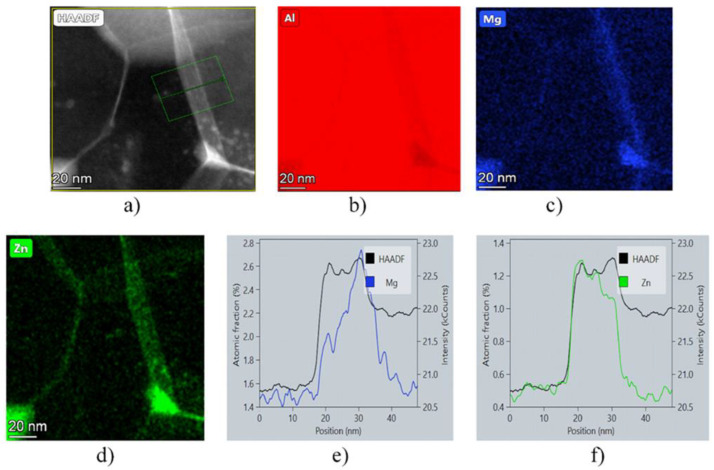
Typical grain boundary of HPT-10 sample shown by (**a**) HAADF image indicating Zn-rich (brightly imaged) boundaries; (**b**–**d**) corresponding element maps for Al, Mg, and Zn, respectively; (**e**,**f**) EDS profile analysis across a boundary, revealing the segregation of Zn and Mg solute atoms into grain boundaries.

**Figure 6 materials-17-00585-f006:**
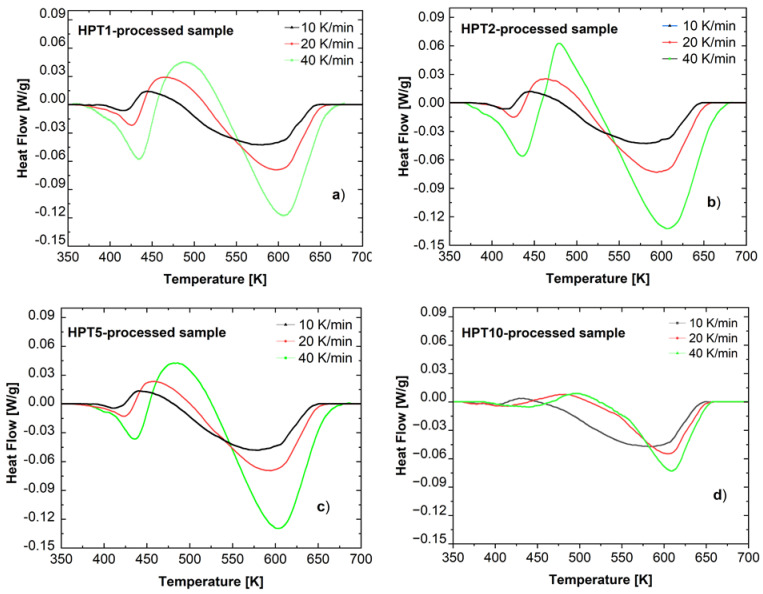
Typical DSC thermograms taken on (**a**) HPT-1, (**b**) HPT-2, (**c**) HPT-5, and (**d**) HPT-10 samples at different heating rates.

**Figure 7 materials-17-00585-f007:**
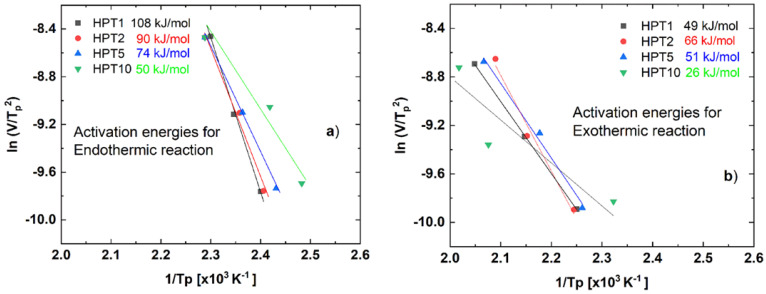
Kissinger plots obtained for (**a**) the first endothermic (dissolution) and (**b**) exothermic (precipitation) reactions in the HPT-processed AlZnMgZr samples.

**Figure 8 materials-17-00585-f008:**
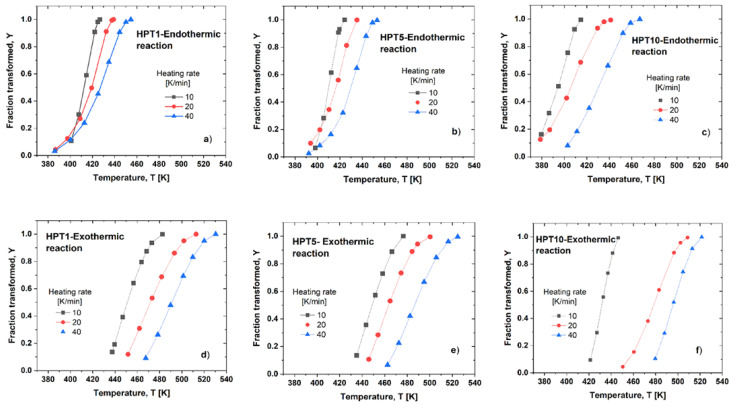
The *Y–T* connections obtained at different heating rates, describing (**a**–**c**) the dissolution and (**d**–**f**) the precipitation reactions in the HPT-1, HPT-5, and HPT-10 samples, respectively.

**Figure 9 materials-17-00585-f009:**
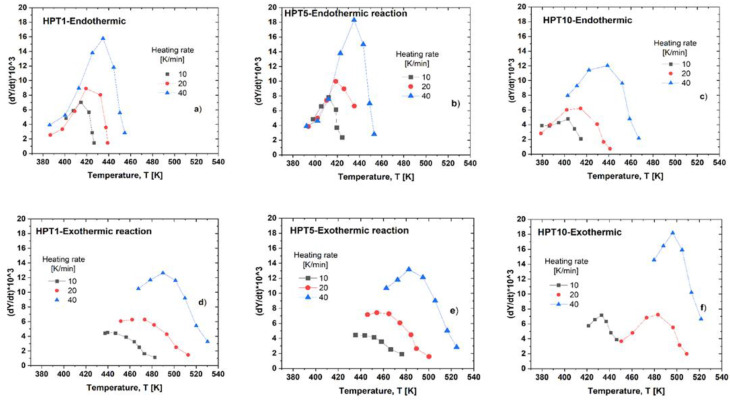
The *dY/dt–T* connections obtained at different heating rates, describing (**a**–**c**) the dissolution and (**d**–**f**) the precipitation in the HPT-1, HPT-5, and HPT-10 samples, respectively.

**Figure 10 materials-17-00585-f010:**
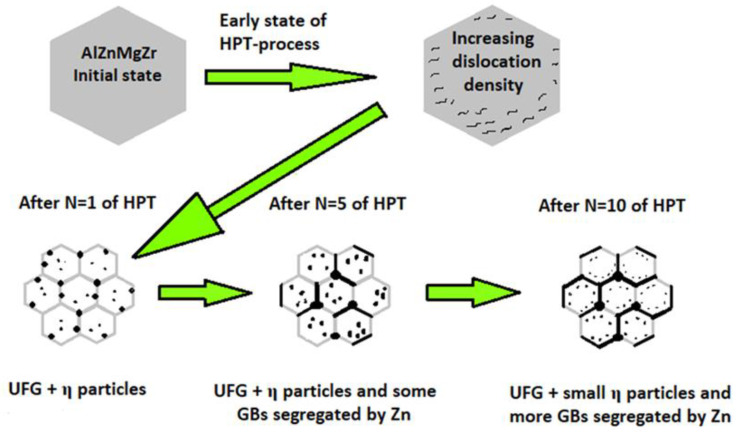
Schematic sequence of the development of the microstructure in 7xxx Al-4.8Zn-1.2Mg-0.14Zr alloy during the HPT process, based on the results of mechanical, TEM, and DSC measurements.

**Table 1 materials-17-00585-t001:** Specific enthalpy values, ∆*H_d_* and ∆*H_p_* obtained for the different HPT-processed Al-Zn-Mg-Zr samples (the values are within 15% relative error).

Reaction Type	HPT-1 Sample	HPT-2 Sample	HPT-5 Sample	HPT-10 Sample
Dissolution (endothermic), ∆*H_d_* (J/g)	2.06	1.83	1.16	0.504
Precipitation (exothermic), ∆*H_p_* (J/g)	3.64	3.04	3.02	0.65

## Data Availability

The raw and processed data required to reproduce these results are available upon reasonable request.

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
