# Peer review of "Microstructure Evolution during High-Pressure Torsion in a 7xxx AlZnMgZr Alloy"

_materials, 2024, doi:10.3390/ma17030585_

Round 1

Reviewer 1 Report

Comments and Suggestions for Authors

1. Page 2, lines 64-66: “Despite extensive investigations, the effect of the different shear strains imposed in HPT processes on same the microstructure and on the subsequent mechanical properties 65 of several Al-Zn-Mg alloys has not been clarified in detail.” 

The mentioned “investigations” should be referenced. What exactly “has not been clarified in detail”? This issue should be clarified in order to substantiate the need for the present work.

2. In the text, the abbreviations ((DSI), (TEM), (DSC)) should be given just one time (see page 2, lines 67-70 and 83-84).

3. The terms “revolutions” and “evolutions” are met in the text and Fig.1. Do they have the same meaning? Please unify them or give explanations for the term “evolutions”.

4. How were the samples for the TEM study prepared?

5. Please give the confidence intervals for the hardness data presented in Fig. 1.

6. What is “N” in Fig.2? Denote it in the footnote.

7. Where are the results of tensile testing? Please add them.

8. Please unify the hardness units (GPa (Fig.1) or MPa).

9. The authors concluded the formation of η-phase (MgZn2 particles). In the article, there is no any evidence (XRD, TEM/SAED) confirming the presence of this phase.

Comments on the Quality of English Language

Minor editing of English language required

Reviewer 2 Report

Comments and Suggestions for Authors

The reference review contains 19 articles up to 10 years and 13 articles up to 5 years out of the 36 items of literature listed. Hence, the conclusion is that the literature review needs to be sufficiently prepared. We recommend extending the list of literature items.
Figure 7 shows changes in activation energies using linear interpolation. That is too coarse an approximation. We recommend improving the figure using another interpolation, e.g. polynomial interpolation (e.g. quadratic function or splines).

Detailed comments:
Line 159 - Figures 3 a and b have very low resolution and are blurred. We recommend a necessary improvement in the quality of the drawings.
Line 198 - Figure 5 - We recommend improving the quality of the images - blurred images, making interpretation of the results difficult.
Line 252 and 253 - In all cases, the 252 data points can be fitted well with a linear line. - This is not true for the HPT10 data (in Figures a and b) - earlier comment about interpolation of data. 
Line 277 - Were HPT11 samples tested ? Not mentioned earlier.
Line 343 - Figure 10 - Is there a logical relationship to allow such an arbitrary interpretation of the microstructure transformation. Figure should be annotated, redrawn or removed. 
Line 399 - Unnecessary text. Text Author 1, A.B.; Author 2, C.D. Title of the article. Abbreviated Journal Name Year, Volume, page range.

  Comments on the Quality of English Language

The article requires minor revisions to improve the clarity and expressiveness of certain text parts.

Reviewer 3 Report

Comments and Suggestions for Authors

The paper describes the evolution of microstructure and tensile properties as a function of revolutions of HPT of a 7xxx series Al alloy. TEM results have been published before, and they are now complemented by DSC. The paper is clear, and I only have a few minor comments.

1. Abbreviations should only be introduced one time only (plus in the abstract), the first time they are mentioned. They should then be used in the text. There are many places where this is not followed.

2. In Figure 1, I assume the shear should be in % (right y-axis).

3. Figure 5; the EDS profile is across a GB, not along.

4. Figure 7a. kj should be kJ.

5. Line 277, should be HPT-10, not HPT-11.

6. Figure 8 and 9. It should be K/min non k/min.

Comments on the Quality of English Language

There are a few sentences/expressions that need attention;

1. Line 65. "on same the microstructure", must be wrong.

2. Line 103-105. This sentence is not complete.

3. Line 214. "we are primarily investigated", must be wrong.

4. Line 282. There is a ")" that should not be there.

There are probably a number of minor grammar issues/typos. Go through the text once more.

Reviewer 4 Report

Comments and Suggestions for Authors

Review for Article-

materials-2816224

Microstructure Evolution during High-Pressure Torsion in a 7xxx Series Al-Zn-Mg-Zr Alloy

This work presented an interesting work about the microstructure study of newly designed 7xxx Al alloys by some special processing methods. Overall, this work is in good shape and high quality. The study of these new approaches and surface characterization methods are of great importance in current studies. The research design and discussions are decent. Personally, this work is of interest to the field study and could be published only after a minor revision. Nonetheless, the manuscript could be improved if the authors could address the comments and recommendations I listed below.

The purpose and novelty of your work should be included in the abstract part. 

In the Introduction part, the authors present a good background and mechanism discussion about SPD and HPT. However, the possible industry applications of your newly designed 7XXX Al alloys are missing. You may need to add some application background about 7XXX Al alloys.(eg. 10.3390/jmse10060740)

Line 78: space between "or" and "10".

More detailed parameters about STEM should be added (eg. ?kV).

The study of intermetallic particles (IMPs) is critical for the Al alloys. The authors should pay more attention to the analysis of IMPs. 

I noticed that Figure 4 is another work figure that may not be directly applied to your work. You may need to conduct another EDS experiment to study the IMPs.  

Comments on the Quality of English Language

 Minor editing of English language required

Round 2

Reviewer 1 Report

Comments and Suggestions for Authors

The paper can be accepted in revised form.